# Diagnosis of Inherited Platelet Disorders on a Blood Smear

**DOI:** 10.3390/jcm9020539

**Published:** 2020-02-17

**Authors:** Carlo Zaninetti, Andreas Greinacher

**Affiliations:** 1Institut für Immunologie und Transfusionsmedizin, Universitätsmedizin Greifswald, 17489 Greifswald, Germany; carlo.zaninetti@uni-greifswald.de; 2University of Pavia, and IRCCS Policlinico San Matteo Foundation, 27100 Pavia, Italy; 3PhD Program of Experimental Medicine, University of Pavia, 27100 Pavia, Italy

**Keywords:** inherited platelet disorders, hereditary thrombocytopenias, blood smear, immunofluorescence, bleeding tendency

## Abstract

Inherited platelet disorders (IPDs) are rare diseases featured by low platelet count and defective platelet function. Patients have variable bleeding diathesis and sometimes additional features that can be congenital or acquired. Identification of an IPD is desirable to avoid misdiagnosis of immune thrombocytopenia and the use of improper treatments. Diagnostic tools include platelet function studies and genetic testing. The latter can be challenging as the correlation of its outcomes with phenotype is not easy. The immune-morphological evaluation of blood smears (by light- and immunofluorescence microscopy) represents a reliable method to phenotype subjects with suspected IPD. It is relatively cheap, not excessively time-consuming and applicable to shipped samples. In some forms, it can provide a diagnosis by itself, as for *MYH9*-RD, or in addition to other first-line tests as aggregometry or flow cytometry. In regard to genetic testing, it can guide specific sequencing. Since only minimal amounts of blood are needed for the preparation of blood smears, it can be used to characterize thrombocytopenia in pediatric patients and even newborns further. In principle, it is based on visualizing alterations in the distribution of proteins, which result from specific genetic mutations by using monoclonal antibodies. It can be applied to identify deficiencies in membrane proteins, disturbed distribution of cytoskeletal proteins, and alpha as well as delta granules. On the other hand, mutations associated with impaired signal transduction are difficult to identify by immunofluorescence of blood smears. This review summarizes technical aspects and the main diagnostic patterns achievable by this method.

## 1. Introduction

### 1.1. Inherited Platelet Disorders

Inherited platelet disorders (IPDs) are a group of rare diseases characterized by reduced platelet numbers and impaired platelet function, causing variable bleeding diathesis. They can affect only platelets, present with other congenital features (syndromic forms), or confer the risk to develop over time additional manifestations such as nephropathy or hematological malignancies (predisposing forms) [1].

The landscape of IPDs comprises, in early 2020, more than 30 disorders and 50 genes. Particularly with the development of high-throughput sequencing tools, the list of IPDs has become enriched rather rapidly [2,3,4,5,6,7,8].

In IPDs, at least one of the three main phases of platelet biogenesis is impaired [9]. The genetic defect can alter the differentiation of hematopoietic stem cells into megakaryocytes (Mks), as in congenital amegakaryocytic thrombocytopenia (CAMT), due to biallelic mutations in the gene coding of the receptor for thrombopoietin (TPO) MPL causing major deficiency or even absence of Mks in the bone marrow [10]. Mutations in transcriptional regulators, such as RUNX1 or ETV6, hit Mk maturation and perturb hematopoietic precursors leading to thrombocytopenia, impaired platelet function, and increased risk of clonal hematopoiesis [11,12,13,14]. Alterations in cytoskeletal proteins, such as non-muscular myosin IIA, β1-tubulin, alpha (α) 1-actinin or filamin A, target the phase of platelet formation and release into blood vessels. In these forms, platelets are typically reduced in number and increased in size [15,16,17,18]. Finally, some IPDs induce thrombocytopenia by reducing the platelet life span, such as changes in glycosylation due to a mutated glucosamine (UDP-*N*-acetyl)-2-epimerase/*N*-acetylmannosamine kinase (*GNE*) gene; or impaired function of the ubiquitination, proteasome system in *ANKRD26*-related thrombocytopenia (*ANKRD26*-RT) [19,20,21,22].

### 1.2. Bleeding Symptoms do not Allow Specific Diagnosis of an IPD

From a clinical point of view, IPDs are characterized by highly variable bleeding tendency. The hemorrhagic history of patients with platelet function defects is significantly more severe than that of patients with thrombocytopenia only, who sometimes present with no bleeding at all. Most affected individuals do not have a spontaneous hemorrhage, but may develop bleeding complications in case of hemostatic challenges such as traumas or surgery. When present, spontaneous bleedings are prevalently muco-cutaneous (e.g., easy bruising, epistaxis, gum bleeding, menorrhagia, gastrointestinal bleeding) and rarely serious [23]. When a hemorrhagic phenotype is overt, the kind of symptoms may help to differentiate “IPD-” from “non-IPD-bleeding-subjects” (e.g., having coagulopathies). In contrast, this criterion is fairly useless in distinguishing among diverse IPDs as symptomatic patients commonly share prevalently mucocutaneous hemorrhages [23].

Apart from a few exceptions, the platelet count in IPDs is stable during life, unless additional causes for thrombocytopenia are acquired. In the absence of compromised platelet function, the severity of bleeding tendency primarily depends on the platelet count [1]. Lifelong bleeding history, female sex, and some types of interventions (e.g., cardiovascular or urological procedures associate with a higher risk of hemorrhagic complications) predict the individual risk of bleeding at surgery [24].

### 1.3. Non-Platelet Features Clinically Characterize some IPDs

Some IPDs present with congenital defects depicting peculiar syndromic pictures. For instance, skeletal defects lead to pathognomonic deformations in Thrombocytopenia-absent radius syndrome (TAR) or Radioulnar synostosis with amegakaryocytic thrombocytopenia (RUSAT) [25,26]. Eczema and mild-to-severe immunodeficiency associate with Wiskott–Aldrich syndrome (WAS) and X-linked thrombocytopenia (XLT), respectively [27,28]. Red blood cell (RBC) thalassemic-like abnormalities are typically found in *GATA1*-related thrombocytopenia (*GATA1*-RT) [29], while Stormorken- and York platelet syndrome share the finding of congenital myopathy [30,31].

“Predisposing IPDs” make patients prone to develop, during life, additional manifestations hitting different systems. With regard to the most prevalent *MYH9*-related disease (*MYH9*-RD), the affected subjects can manifest sensorineural deafness, nephropathy, juvenile cataracts, and elevation of liver enzymes. The individual risk of such features depends on the location of the mutation, which differently affects protein function [32]. A risk for deafness is also present in patients with macrothrombocytopenia due to *DIAPH1* mutations [33].

Much more relevant for prognosis, all patients with CAMT develop critical bone-marrow aplasia during infancy [34]. A similar evolution can affect the proportion of patients with RUSAT [26]. A possible progression to hematological malignancies has been reported in three forms of autosomal dominant (AD) IPDs with congenital thrombocytopenia with normal-size platelets [35,36]. The proportion of patients developing myelogenous or lymphoblastic leukemia, or myelodysplastic syndrome, ranges from less than 10% in *ANKRD26*-RT to about 25% and 33% in *ETV6*-related thrombocytopenia (*ETV6*-RT) and in Familial platelet disorder with propensity to acute myelogenous leukemia (FDP/AML), respectively [12,13,37,38,39,40,41,42]. A potential onset of juvenile myelofibrosis, splenomegaly, and osteoporosis has been reported in subjects with mutations in tyrosine kinase SRC (*SRC*-related thrombocytopenia) [43].

### 1.4. Treatment of IPDs is Usually Symptomatic

Hematopoietic stem cell transplantation (HSCT) is indicated for CAMT, WAS, severe phenotypes, or poor prognosis IPDs; otherwise, treatment is mostly symptomatic [44,45]. Spontaneous and provoked bleedings can be prevented by simple behavioral norms such as avoiding antiplatelet drugs and contact sports, providing regular dental care, and reducing menorrhagia with hormonal therapy. For the therapeutic management of moderate hemorrhage, anti-fibrinolytic agents and desmopressin are effective options. Only in the case of severe bleeding, platelet concentrates, or pro-hemostatic compounds, such as recombinant activated factor VII (rFVIIa) are indicated [46,47,48]. In preparation for surgical procedures, the increase of platelet number and/or the restoring of platelet function can be required. Especially in IPDs with defective platelet function, anti-hemorrhagic prophylaxis is associated with a significant reduction of bleeding frequency [24]. The threshold platelet levels considered safe to undergo invasive procedures vary according to the kind of hemostatic challenge (e.g., childbirth, or major surgery), and the severity of platelet dysfunction. In the case of a major hemostatic challenge in a patient with major platelet dysfunction, platelet transfusion is recommended independently of the platelet count target [49,50,51]. However, especially in Glanzmann’s thrombasthenia (GT), platelet transfusion should be restricted to situations in which other treatments fail to reduce the risk of alloimmunization of patients against the glycoprotein lacking on their own platelets. To transiently increase the platelet count, a thrombopoietin-receptor agonist (TPO-RA) may represent an alternative to platelet concentrates in a subgroup of IPDs [52,53]. Only in WAS, splenectomy represents an option to increase platelet count. Splenectomy should not, however, be adopted in patients who are candidates for HSCT as it worsens the disease-related immunodeficiency [54]. Valid alternatives for these subjects are gene therapy, or TPO-RAs [55,56]. Effective strategies to cure extra-hematological features of *MYH9*-RD, such as renin-angiotensin pathway inhibitors to slow nephropathy progression, or cochlear implantation to overcome profound deafness, have become available during the last decade for patients at risk for these complications [57,58].

### 1.5. Current Diagnostic Tools for IPDs

IPDs are suspected when there is an unusual bleeding history and/or other phenotypic clues are present. When other affected relatives are present, or peculiar syndromic features are detectable, the diagnosis of an IPD can be made relatively easily. However, identification of the altered platelet structure is usually very challenging. On the other hand, the lack of inheritance trait does not exclude IPDs. Indeed, sporadic cases by de novo mutations or mosaicism have been reported in more than one-third of patients with *MYH9*-RD, one of the less rare IPDs [59].

Besides clinical evaluation, including the documentation of hemorrhagic history using a validated assessment tool, such as the ISTH Bleeding Assessment Tool (ISTH-BAT) [23,60], the first diagnostic step for IPDs requires an estimation of platelet size [61], light-transmission aggregometry (along with the evaluation of dense granule release by lumiaggregometry), and flow cytometry. Generally, only selected patients undergo, in the first instance, further sophisticated investigations like western blotting, electron microscopy, or the measurement of certain enzyme activity [62]. These platelet function assays have several shortcomings. Globally, the level of standardization of such assays is low. In addition, relatively large amounts of fresh blood are required, which makes it impossible to ship samples, and requires patients to be at the place of analysis. For aggregometry, a minimum of platelet concentration (i.e., not less than 80 × 10^3^ per µl) is required within platelet-rich plasma or whole blood for effective analysis [63,64,65]. This can pose a challenge when dealing with patients having moderate or severe thrombocytopenia. Additionally, the collection of large volumes of blood is not possible from neonates and young children.

Genetic analysis has taken a big leap forward during the last ten years. The spread of next-generation sequencing (NGS) greatly contributed to the unraveling of the genetic basis of several forms of IPD, and thus immediately became a significant diagnostic tool in the field. Using a small amount of blood with minor risk of pre-analytical artifacts, it can provide patients with a highly accurate diagnosis [4,5,6,7,66,67,68,69,70,71,72]. As it becomes cheaper to perform, it is not inconceivable that NGS could become the standard above other diagnostic techniques. However, interpretation of genetic findings can be difficult especially in patients without a peculiar clinical picture. In fact, NGS usually detects many variants of unknown or uncertain significance, whose correlation with the patient’s phenotype is unresolved, and over-interpretation is a major risk [73,74,75,76,77]. Data from big case series reported so far indicated that NGS is able to achieve a diagnosis in nearly 50% of IPD patients suffering from congenital thrombocytopenia, and in nearly 25% of those with inherited platelet function disorder. Taken together, these data show that the probability of achieving a definite diagnosis by NGS is very high in selected and well-characterized individuals, but the underlying molecular cause of an IPD may still not be identified in a large number of patients.

### 1.6. Main Advantages of Blood Smear Analysis

In such a landscape, the immune-morphological analysis of platelets on the blood smear may represent a key approach in the work-up of IPDs both for feasibility and diagnostic power for the following reasons: first, it requires a low volume of blood (less than 100 µl), thus is easily collected even from newborns [78]. Secondly, samples can be shipped safely, even long distances, to reach the diagnostic reference centers. Thirdly, although requiring specialized expertise, it is relatively cheap and not excessively time-consuming. For instance, at our Institute, the overall procedure (including staining and assessing the blood smears for a large panel of markers) takes about 45 min per patient. Fourthly, the immune-morphological evaluation of blood smears allows more detailed phenotyping of the platelets [79]. The double-analysis approach (i.e., by light- and immunofluorescence-microscope, see below) can provide diagnosis by itself, as in the case of *MYH9*-RD, or in addition to other first-line tests such as aggregometry and flow cytometry as for GT and its variants (type 2- and type 3 GT), or for classical and monoallelic Bernard–Soulier syndrome (BSS) [80,81,82,83]. In addition, it can indicate for clinicians that the patient may have an increased risk for bleeding during interventions, as in the case of major alterations in α- or dense granule content (i.e., Grey Platelet Syndrome, GPS, α-,δ-, or α+δ storage pool disease, SPD) [62,84]. By labeling a few proteins, a potential abnormality could be identified or narrowed-down to a group of IPDs featured by cytoskeleton alterations (e.g., *FLNA*-related thrombocytopenia, *FLNA*-RT [17], *TUBB1*-related thrombocytopenia, *TUBB1*-RT [85], and WAS). Likewise, it is possible to differentiate between GPS and *GFI1B*-related thrombocytopenia (*GFI1B*-RT), in the case of large and pale staining platelets in the May-Grünwald Giemsa (MGG) blood smear. Both have major deficiencies in α-granule proteins, but only *GFI1B*-RT platelets express the stem cell antigen CD34 [86].

An international study that involved 3217 patients referred to four major specialized centers demonstrated that double-stained blood smear analysis was able to achieve a diagnosis of IPD (confirmed by parallel genetic testing and standard laboratory tools) in 27% of the referred subjects. This proportion was equivalent to those achieved in the major previous case series, where all the available diagnostic tools, including those requiring a larger amount of blood, were exploited [78].

## 2. Technical Considerations

### 2.1. Preparation of Blood Smears

Either capillary blood, obtained by finger-prick, or anticoagulated blood can be used for the preparation of blood smears. In the latter case, the choice of ethylenediaminetetraacetic acid (EDTA) is advisable with respect to other anticoagulants (e.g., citrate, or hirudin) since it proved to maintain superior stability of immune-morphological markers [87]. When using anticoagulated blood, slides should be prepared as soon as possible (within 4 h) after drawing to avoid the increase of platelet size due to low calcium concentration. Since the marginal band of microtubules depolymerizes at low temperatures, leading to alteration of the discoid platelet shape, the blood should be stored at temperatures > 20 °C [88].

The blood smear preparation is relatively simple. When correctly executed, blood films are stable for several days and can be shipped by regular mail even over long distances. The slides should be preferably prepared by skilled persons, by following a few simple recommendations. Glass slides have to be clean and fat-free. An about 3–4 µL blood drop is placed on the base-slide 1 cm from the edge. The spreader-slide, inclined at an angle of about 40 degrees, is then placed on the base slide, ahead of the drop, and pulled back to make contact with blood. As soon as the drop has spread out along the line of contact, the smear is obtained by rapidly moving towards the spreader-slide. The ideal length of the smear is 2/3 of that of the base-slide. To form a cell-monolayer, the ideal smear has to be neither too thin nor too thick. Thickness depends on the dimension of the blood drop, the angle of the spreader-slide, and the speed of the spreading. The lateral edges and an oval-shaped tail should be visible (Figure 1). Once smeared, the slides should be immediately air-dried [89]. A You-Tube instruction on how to make the blood smears is available under the following link: https://www.youtube.com/watch?v=BbBcWp5LAXs.

Artefactual alterations of platelets or leukocytes on the smear can be induced by hemolysis due to moisture precipitation occurring during air freight when a cool glass slide comes into contact with warm, humid air. Hence, it is recommended to wrap the slides before shipment.

At least 15 to 20 slides should be collected from each patient, as typically only two colors per slide (red and green = two different antibodies) are used for immunofluorescence staining.

### 2.2. Immunofluorescence Labelling

Immunofluorescence staining and analysis are usually performed in specialized laboratories. Nevertheless, referring physicians should take into account that slides should ideally be fixed within three days after film preparation, and the timing of shipment has to be planned accordingly.

Before immunofluorescence staining, slides undergo fixation using different methods according to the protocol in use in the laboratory. The main options are ice-cold acetone (−20 °C; 2–5 min, permeabilizing), or methanol (−20 °C, 1 min, permeabilizing), or 1–3% paraformaldehyde (room temperature for 10–20 min, followed by permeabilization with 0.1–0.2% Triton X-100 for 5–10 min). After fixation and permeabilization, blood smears are covered with monoclonal or polyclonal antibodies against the target proteins [78]. Fluorescence can be obtained by antibodies directly conjugated with a fluorescent dye, or by secondary antibodies labeled with fluorescent dyes that bind to primary antibodies. By staining in parallel the target protein (e.g., MYH9) and another platelet-specific marker (e.g., β1-tubulin), easy identification of the distribution of the protein of interest can be achieved. To counterstain nuclei, Hoechst or 4′, 6-diamidino-2-phenylindole (DAPI) are commonly used. It is highly advisable to always stain for two platelet structures. Otherwise, the interpretation becomes very difficult if a protein is missing. The expression pattern of the target proteins is then detected by comparison with a normal control smear stained in parallel.

### 2.3. Note of Caution

Although binding to the same protein, different monoclonal antibodies differ in their avidity and binding strength. Especially for quantitative assessment, appropriate controls and quality measures are mandatory. In addition, the absence of binding of a monoclonal antibody does not prove the absence of the protein. In the case of a polymorphism of the gene affecting only the binding site of the monoclonal antibody, the protein may be present despite the antibody, not binding.

## 3. Main Diagnostic Patterns

### 3.1. Light Microscopy

MGG- or Wright-stained blood film analysis represents the first step of the differential diagnosis of IPDs on a blood smear. First, it allows an estimate of the platelet count, which can be underestimated by automated cell counters in patients with platelet macrocytosis [90]. Second, it enables the evaluation of platelet size and platelet staining. In addition, light microscopy can identify morphological abnormalities of other blood cells, which can guide the diagnostic work-up. Conventionally, at least 100 platelets (and other blood cells) per smear should be observed.

### 3.2. Platelet Size

The assessment of the mean platelet diameter (MPD) is a key step of the differential diagnosis of IPDs. It can be achieved by an ocular micrometer or by software-assisted image analysis. A collaborative study on a large series of subjects with 19 different disorders proposed a classification of IPDs according to platelet diameter. Namely, it distinguished forms with enlarged (MPD > 3.2 µm), decreased (MPD < 2.6 µm), and normal (2.6 µm < MPD < 3.2 µm) MPD [61]. Particularly, the finding of MPD > 3.9 µm (corresponding to about half the diameter of a normal red blood cell) or < 2.6 µm showed good sensitivity and specificity in differentiating IPDs with giant platelets (i.e., *MYH9*-RD and BSS) from those with small platelets (i.e., TAR, CAMT, and WAS/XLT), respectively. An example of normal and altered platelet dimensions is shown in Figure 2, Panel I.

### 3.3. Platelet Staining

The presence of azurophilic granules (e.g., platelet α-granules) determines the typical eosinophilic aspect of platelets. A reduction of platelet staining can suggest at least three forms of IPDs with thrombocytopenia and normal-to-enlarged platelet size. In GPS, the prototypical form of this group, α-granules are almost completely absent, and platelets appear large and empty [84]. In *GFI1B*-RT or *GATA1*-RT, a partial reduction of granularity making platelets “pale” can be detected [91,92]. Regarding α-, δ- or α+δ SPD, only in the case of a severe reduction of α-granules will platelets appear less stained by light microscopy. An example of normal and reduced platelet staining is shown in Figure 2, Panel II.

### 3.4. Other Morphological Alterations of Platelets

The presence of platelet clumps (Figure 2, Panel II), despite the best efforts to avoid platelet activation during blood-letting and preparation of blood smears, can suggest type 2B- or platelet-type von Willebrand disease (vWD) [93,94]. In the latter, platelets are often enlarged [95].

The presence of giant α-granules can indicate Thrombocytopenia Paris-Trousseau (TCPT) [96]. Prominent vacuoles can be observed in the platelets of patients affected by *GATA1*-RT [97].

*Note of caution*: the most frequent reasons for platelet aggregates on the blood smear is the use of blood from a finger prick or the time taken between dropping the 3 µL of blood onto the slide and creating the smear was too long resulting in the activation of platelets through contact with the glass surface.

## 4. Morphological Alterations of Other Blood Cells

When dealing with large platelets, the “reader” of the blood smear should carefully look for possible basophil inclusions in neutrophils’ cytoplasm, which are pathognomonic in *MYH9*-RD [96]. It has to be taken into account that the so-called “Döhle-like” bodies, always identifiable by immunofluorescence, are detectable only in a proportion of the affected patients by light microscopy (see below).

Among subjects with pale platelets, the finding of red blood cell (RBC) anisopoichilocytosis (i.e., cells having prominent differences in size and shape) can corroborate the suspicion of *GATA1*-RT [97]. The presence of erythrocytes resembling a mouth (i.e., stomatocytes) plus platelet macrocytosis can indicate sitosterolemia (*STSL*-RT) [98].

In regard to RBCs, microcytosis and hypochromia are often detectable in symptomatic IPD patients, independently on their specific form, because of bleeding-related iron deficiency anemia.

### 4.1. Immunofluorescence Microscopy

Immunofluorescence analysis represents the second step of the differential diagnosis of IPDs on a blood smear. Several platelet proteins, belonging to different categories (see below) can be labeled by antibodies. Based on clinical features (e.g., hereditary trait, presence of syndromic picture) or morphological aspects (e.g., enlarged or small platelets), a proper panel of antigens to be tested can be defined. Another approach is to use a standard minimum-panel for all the patients under investigation, to be probably integrated as a second step with further staining or other diagnostic tests. This latter approach allows us to gain more information and to provide a guidance answer more rapidly. Most important, it allows the blood smear reader to gain experience of how certain platelet structures are altered in various IPDs.

### 4.2. Cytoskeleton Proteins

A considerable group of IPDs are characterized by alterations in cytoskeleton proteins, reducing platelet production, and altering platelet size.

The platelet cytoskeleton often shows alterations in various IPDs associated with platelet macrocytosis as well as in acquired platelet disorders. In addition, it can be an artifact in case of platelet pre-activation or cold storage of the blood tube before preparing the smear.

Abnormal distribution of the *MYH9* coded protein non-muscular myosin IIA heavy chain in the cytoplasm of neutrophils drives the diagnosis of *MYH9*-RD with high sensitivity and specificity [96]. The pathognomonic inclusions, not always detectable in the MGG-stained blood smear, are usually evident in immunofluorescence (Figure 3, Panel I). At least two main patterns have been described. The one is typical for mutations involving the tail domain of non-muscular myosin IIA and consists of one-to-three large (2–5 μm) aggregates per cell together with further small aggregates. Conversely, mutations involving the head domain of the protein produce several smaller (<0.5 µm) aggregates. Not all leukocytes may express these inclusion bodies, especially in the case of a genetic mosaic of the patient.

Double labeling for cytoskeleton markers β1-tubulin and filamin A enables to recognize patterns suggestive for two forms of IPD with large, normal-stained platelets: *TUBB1*-RT and *FLNA*-RT [17,85]. *TUBB1*-RT is characterized by the loss of the typical peripheral ring of β1-tubulin, which forms images similar to balls of yarn (Figure 3, Panel II).

In *FLNA*-RT, up to one-half of platelets are lacking filamin A staining, while displaying the normal β1-tubulin distribution. Another typical feature of *FLNA*-RT we have observed in our laboratory is diffuse staining of platelets for filamin A, which usually appears as a thick ring (Figure 3, Panel III).

The use of an antibody against β1-tubulin is also worthwhile to identify some patients with WAS [78]. In this disorder, platelets are reduced in size and may show the abnormal distribution of this cytoskeleton protein that appears stretched or forming uncommon images (e.g., eight-shaped aspect) (Figure 3, Panel IV). However, not all WAS patients show these changes in β1-tubulin, and a normal β1-tubulin ring does not exclude WAS.

### 4.3. Surface Receptors

Two IPDs can be readily identified by labeling proteins belonging to the membrane receptor complex GpIb/IX/V or GpIIb/IIIa [78].

Patients with biallelic BSS display complete or severe depletion of GpIb/IX (Figure 4, Panel I). Monoallelic form of BSS, having a partial reduction of these glycoproteins, can be sometimes suspected by immunofluorescence. However, immunofluorescence analysis on the blood smear is at best semiquantitative, and reduced surface Gp expression requires confirmation by flow cytometry.

Similarly, in “classic” GT platelets are normal in number and size, but the expression of GpIIb/IIIa on the surface is abolished or severely reduced (Figure 4, Panel II). Only in type 3 (i.e., monoallelic), GT platelets are often enlarged. Immunofluorescence can also reveal a partial reduction of GpIIb/IIIa in type 2 or 3 GT (see comment above).

### 4.4. Alpha and Delta Granule Markers

Labeling for von Willebrand factor (vWF), thrombospondin (TS), and P-selectin (P-sel) enables to detect the lack of α-granules, thus confirming GPS [78]. Partial depletion of α-granules can refer to two IPDs with a low number of normal or enlarged platelets: *GATA1*-RT and *GFI1B*-RT. These forms can be further distinguished in the presence of dyserythropoiesis or by the expression of the stem cell antigen CD34 on platelets, respectively [86] (Figure 5). In the absence of thrombocytopenia, a decrease of α-granule markers moves towards α-SPD (Figure 6, Panel I).

A set of dense granule markers as Lamp 1 (L1), which is present also in the lysosomes, Lamp 2 (L2), and CD63, allows identifying forms with δ-granules reduction, δ-SPD (Figure 6, Panel II). If the patient also presents with oculo-cutaneous albinism Hermansky–Pudlak syndrome is the most likely diagnosis. Abnormalities of δ-granule distribution associated with mild thrombocytopenia suggest another IPD, which will be discussed below (i.e., *ETV6*-RT). Finally, it should be recalled that either acquired platelet defects or preanalytical platelet activation can result in alterations of the distribution of granule markers.

*Note of caution*. The main pathologic patterns of IPDs reported in Figure 3, Figure 4, Figure 5 and Figure 6 were obtained by a regular immunofluorescence microscope. By exploiting more sophisticated technologies, such as confocal microscopy, one can achieve higher quality images, as shown, as an example, in Figure 7. It allows us to get better definition and counterstain and to visualize the specific distribution of platelet markers more clearly. In contrast, only a few centers have a confocal microscope available, and the time to obtain the images is longer and not feasible for diagnostic purposes. Since the aim of this review is to guide physicians in the diagnostic work-up of IPDs, we deliberately decided to present primarily images obtained by regular immunofluorescence microscopy. Although the recognition of certain pathologic patterns can be more difficult, it is more representative of the real-life diagnostic setting.

## 5. Weaknesses and Ethical Considerations

### 5.1. Limitations Related to this Method and New Perspectives

The immune-morphological analysis of blood smears needs skilled operators and can be performed only in specialized centers. This logistic limitation can be easily overcome by shipping samples. With a view to the future, the possibility of automatic or semi-automatic reading of the immunofluorescence patterns is attractive as it could mitigate some logistical difficulties. Similar automated readers are established for routine practice in rheumatology laboratories.

Some preanalytical issues are still debated. First, it has not been clarified whether the smears obtained by finger prick can lead to artifacts by inducing platelet activation. Similarly, also EDTA-dependent platelet pre-activation can alter some morphological parameters on slides obtained by anticoagulated blood [99]. Moreover, spontaneous activation can affect platelet markers, mainly cytoskeleton proteins (e.g., due to preanalytic alterations, signaling defects, or IPDs themselves as type 2B vWD or monoallelic GT). None of the morphological markers of platelet activation can differentiate between in-vivo activation and preanalytical arteifacts.

In regard to the diagnostic power, sensitivity, and specificity of the immunofluorescence method seem to be very high in *MYH9*-RD [78,96,100,101]. Similar performance parameters were found for GT and biallelic BSS [78]. Encouraging data have also been reported about the identification capability of GPS by blood smear assessment [84,102]. About the other forms, due to the scarceness of patients, sensitivity and specificity evaluation are not achievable yet.

When all immune-morphological patterns are normal, the present method can provide rapid help to the differential diagnosis by excluding some IPDs. On the other hand, the global negative predictive value is questionable since other acquired or inherited disorders, causing thrombocytopenia and platelet dysfunction, cannot be excluded nor suggested.

To face these limitations, further network studies involving a broader number of patients, and sharing samples among laboratories, should be encouraged.

New substantial diagnostic perspectives rely on the identification of specific changes in platelet proteins being recognizable by diagnostic antibodies. A possible diagnostic pattern for *ACTN1*-related thrombocytopenia has been proposed upon observing the peculiar distribution of MYH9 protein in the granulomere area of surface-activated platelets [103]. Besides, alteration in MYH10 distribution has been reported as a sign of thrombocytopenia due to *FLI1* or *RUNX1* mutations [104]. On the side of platelet function defects, the possibility to mark phosphorylated proteins along the major signaling pathways seems promising to enlarge the diagnostic spectrum of these disorders [43].

### 5.2. Ethical Considerations

Mainly with the increasing use of NGS, some major ethical issues related to IPDs diagnosis have arisen. They prevalently concern forms conferring the risk of acquired hematological malignancies as *ANKRD26*-RT, *ETV6*-RT, and FDP/AML. These disorders are largely characterized by mild thrombocytopenia and minor bleeding tendency. So, at least at the diagnosis stage, their impact on patients’ quality of life can be almost negligible. In contrast, the additional information about an increased risk of leukemia can considerably enhance the psychological disease burden of patients and parents (if the patient is a minor). Notably, the global proportion of patients developing hematological malignancies is low, with 10 to 30% [12,13,35,36,37,38,39,40,41,42]. Moreover, we do not have at now efficient tools to predict the individual risk nor to prevent the development of leukemia.

Currently, experts in the field are discussing the implications of testing for IPDs in regard to IPDs associated with increased risk for hematological malignancies [77,105]. Before undergoing genetic testing, patients are usually requested to give informed consent, and they can opt not to receive additional incidental information (e.g., in a female patient, to be a carrier of hemophilia A) [106]. Conversely, in the IPDs setting of *ANKRD26*-RT, *ETV6*-RT, and FDP/AML, it is not possible to separate the diagnostic- from the predictive value of the mutation. Also, immunofluorescence testing can raise the suspicion at least for *ETV6*-RT. In eight out of nine patients suffering from this form, detectable ETV6 proteins within the platelet cytoplasm (absent in the healthy controls) have been reported. Likewise, dense granule markers L2 and CD63 were diminished and diffusely distributed in *ETV6*-RT platelets [107]. Before including antibodies staining for ETV6 into the panel, specific consent should be obtained from the patient.

## 6. Conclusions

The diagnostic method to identify IPDs using a blood smear presents a helpful “platform of action”. It allows rather extended diagnosis in young children using minimal amounts of blood. Blood smears can be easily prepared even at sites far away from specialized laboratories and can be mailed for further diagnostic workup. The method powerfully suggests diagnosis in some cases (e.g., *MYH9*-RD, GT, and BSS), making it possible to achieve the genetic confirmation restricted to the gene of interest. A recapitulatory, guidance flow-chart for diagnostic testing for IPDs is shown in Figure 8.

Except a few selected forms, the main benefit of a confirmed diagnosis of IPD is to avoid over-treatment. In fact, one of the main risks of these patients is to be misdiagnosed as having immune thrombocytopenia (ITP) or, less frequently, myelodysplastic syndrome. These misdiagnoses seriously impact their management by exposing patients to ineffective and potentially harmful treatments such as splenectomy, immunosuppression, or chemotherapy [32,51,108].

To progressively understand IPDs, further characterization of patients is highly relevant. The combination of genetic testing, together with detailed phenotypic characterization and systems biology studies in the laboratory are essential. Immunofluorescence analysis of platelets on a blood smear is one of the new tools contributing to these joint efforts. Moreover, it allows the rapid translational application of new findings on IPDs obtained by research laboratories using mouse models and cultured Mks into clinical practice.

## Figures and Tables

**Figure 1 jcm-09-00539-f001:**
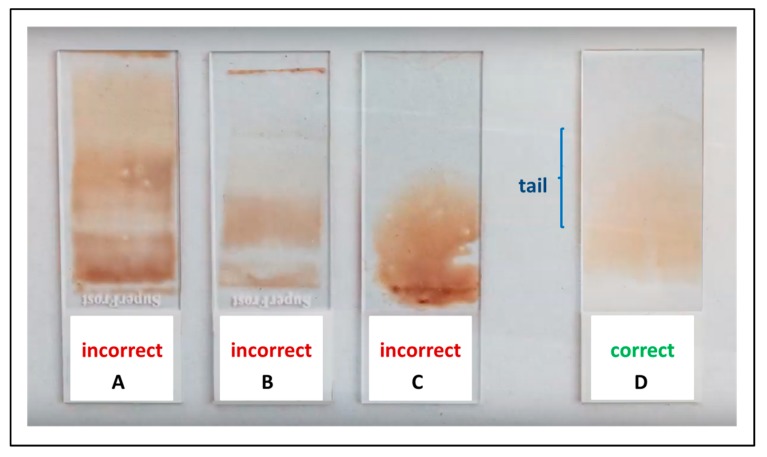
**Blood smear preparation.** (**A**–**C**) Three examples of incorrect blood smears because of excessive thickness of the blood film and the absence of the monolayer tail. (**D**) An ideal blood smear with evident lateral edges and a visible oval-shaped tail.

**Figure 2 jcm-09-00539-f002:**
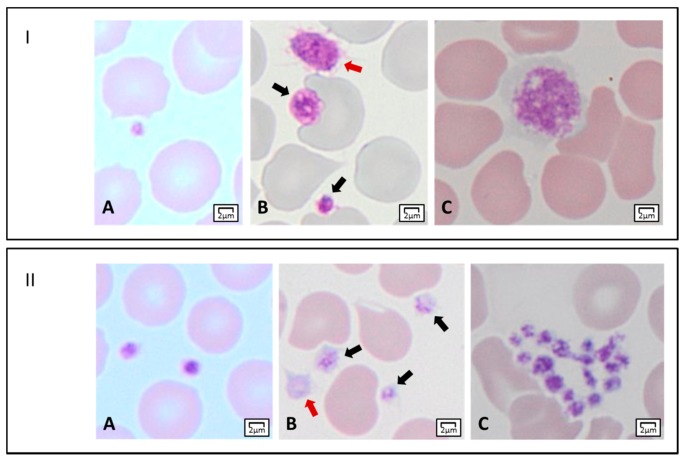
**Light microscopy features of platelets****.** Panel I. Platelet dimensions. (**A**) A small platelet (mean platelet diameter, MPD < 2 µm). (**B**) Two platelets within the normal range of MPD (black arrows), and an enlarged platelet (MPD > 3.5 µm) (red arrow). (**C**) A giant platelet with MPD corresponding to about that of a normal red blood cell. Panel II. Platelet staining, and morphology. (**A**) Normal-stained platelets. (**B**) Platelets with reduced granular labeling (black arrows), and a gray platelet (red arrow). (**C**) A platelet clump.

**Figure 3 jcm-09-00539-f003:**
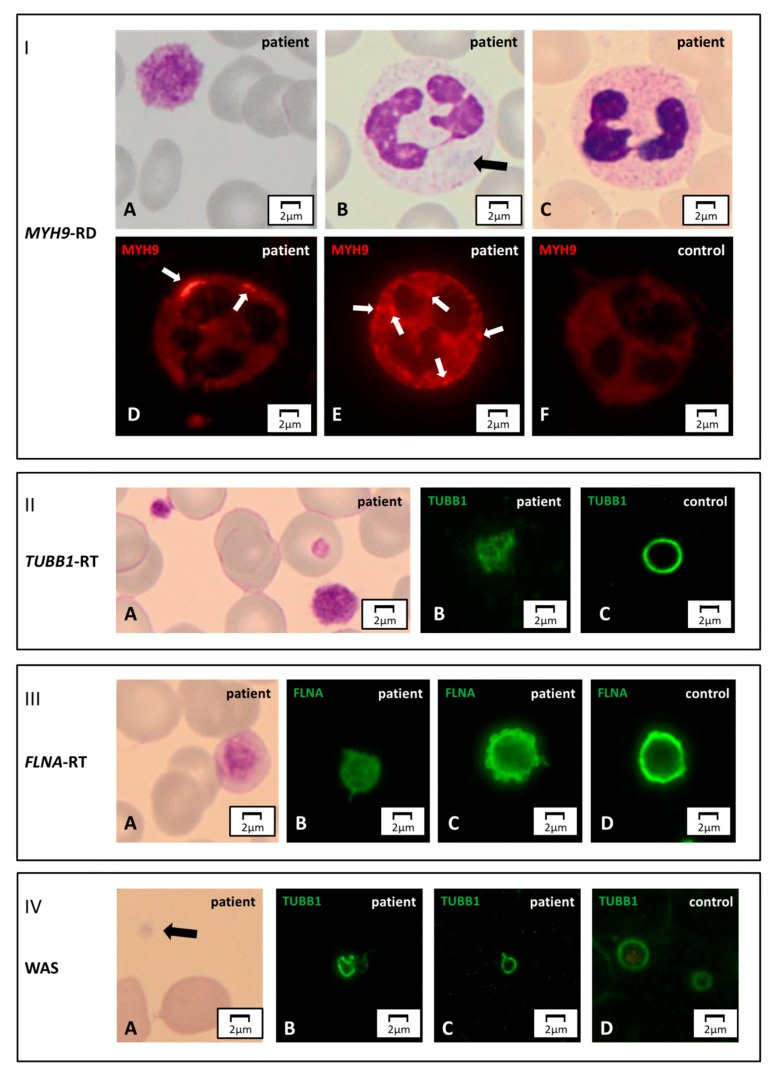
**Cytoskeleton markers.** Panel I. *MYH9*-RD. (**A**) A normal-stained giant platelet and (**B**) a faint blue Döhle-like body in a neutrophil’s cytoplasm of an *MYH9*-RD patient, by light microscopy. (**C**) The aggregates of mutated non-muscular myosin II-A heavy chain can be unrecognizable in May-Grünwald Giemsa (MGG)-stained slides, while they are always detectable by immunofluorescence microscopy. (**F**) In the control, the non-muscular myosin II-A heavy chain signal is diffuse, while in the affected subjects it forms aggregates of different sizes and numbers. (**D**) Few, large (2–5 μm) inclusions are typically observed in patients with mutations hitting the tail domain of *MYH9* coded protein. (**E**) Multiple, smaller (<0.5 µm) aggregates are characteristic of patients carrying mutations in the head domain of the same protein. Panel II. *TUBB1*-related thrombocytopenia (*TUBB1*-RT). (**A**) An MGG-stained blood smear of an affected subject displaying large, normal-stained platelets. (**B**,**C**) In immunofluorescence, the typical β1-tubulin ring (control) is disrupted and resembles a ball of wool. Panel III. *FLNA*-related thrombocytopenia (*FLNA*-RT). A) Patients suffering from *FLNA*-RT have macro-thrombocytopenia with normal granular staining of platelets. (**B**–**D**) By immunofluorescence microscopy, filamin A presents altered distribution with respect to the typical peripheral ring-shaped signal (control). In the affected subjects, the filamin A pattern can appear diffused or patched. Panel IV. WAS. (**A**) Patients distinctively show small platelets in the MGG-stained blood smear. (**B**–**D**) The genetic alterations make platelet cytoskeleton stiffer, thus causing an abnormal distribution of β1-tubulin by immunofluorescence assessment. With respect to the typical ring (control), it can present stretched or bent.

**Figure 4 jcm-09-00539-f004:**
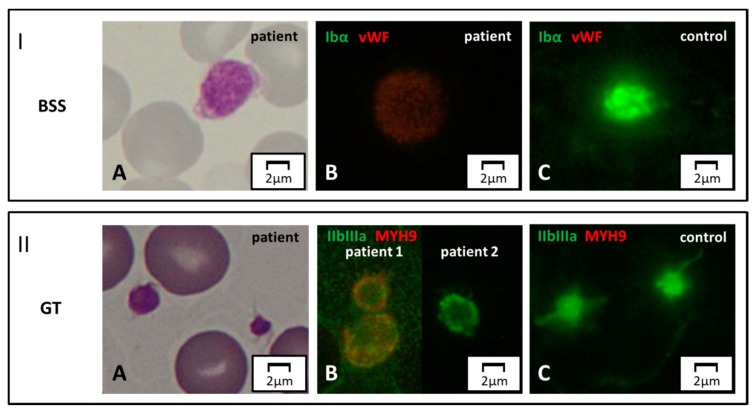
**Surface protein markers.** Panel I. Bernard–Soulier syndrome (BSS). (**A**) Light microscopy. A patient affected with biallelic BSS displays a large platelet on MGG-stained blood smear. (**B, C**) Immunofluorescence microscopy. By labeling for GpIbα (green) and vWF (red), the green signal is completely abolished in the patient due to the absence of the corresponding protein on the surface of platelets. Panel II. GT. (**A**) Normal-size platelets in a subject affected with GT, by light microscopy. (**B, C**) In immunofluorescence, complete and partial reduction of the GpIIb/IIIa signal (green) in two patients with respect to control. Note of caution. Although the immunofluorescence pictures are taken by the green-light channel, a dim red signal emerges in some pathologic samples due to counterstaining for a second marker in red (in this case, MYH9).

**Figure 5 jcm-09-00539-f005:**
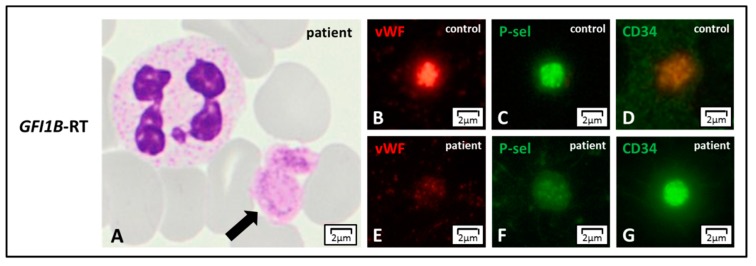
***GFI1B*-RT. (A**) *GFI1B*-related thrombocytopenia (*GFI1B*-RT) is characterized by large and pale platelets, by light microscopy. **(B**–**G**) The reduction of azurophilic granules’ content can be confirmed in immunofluorescence by staining the typical alpha-granule markers such as von Willebrand factor (vWF) or P-selectin (P-sel) that appear reduced with respect to control. Of note, platelets also present CD34 stem cell antigen that is normally not expressed in the controls.

**Figure 6 jcm-09-00539-f006:**
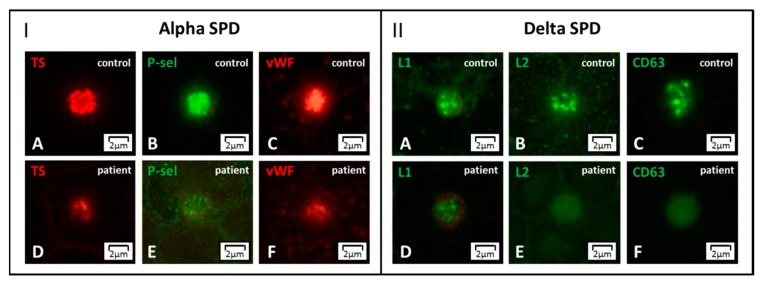
**Alpha- and delta storage pool disease (SPD) by immunofluorescence microscopy.** Panel I. (**A**–**F**) In alpha SPD, the labeling for the alpha-granule markers thrombospondin (TS), P-selectin (P-sel), and von Willebrand factor (vWF) shows significantly reduced staining with respect to control. Panel II. (**A**–**F**) In delta SPD, the dense granule markers Lamp 1 (L1), Lamp 2 (L2), and CD63 display diffuse patterns with reduced or even absent granularity. Alpha- and delta SPD can present associated.

**Figure 7 jcm-09-00539-f007:**
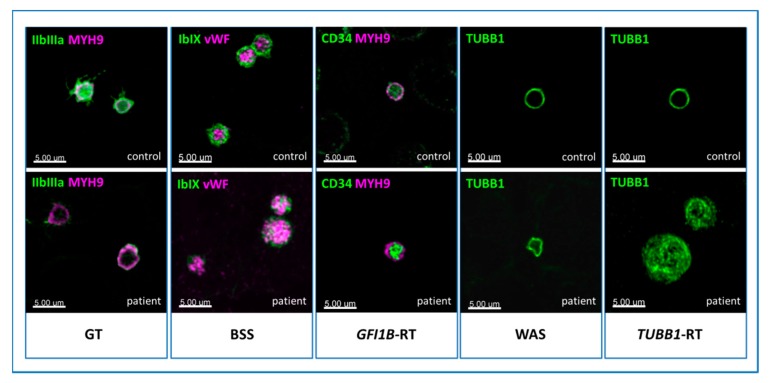
**Images of pathologic patterns of five forms of inherited platelet disorder (IPD) obtained by confocal microscopy.***Immunofluorescence microscopy*. In Glanzmann’s thrombasthenia (GT), platelets show almost no GpIIb/IIIa surface receptor (green). In BSS, the signal corresponding to GpIb/IX (green) on the surface is lacking, and only some spots inside the platelets are recognizable, showing the altered protein which cannot be transferred to the membrane. In *GFI1B*-RT, the stem cell antigen CD34 (green) persists on the platelet surface. Instead of a typical nice ring (controls), the cytoskeleton marker β1-tubulin (green) appears twisted in Wiskott-Aldrich syndrome (WAS), and resembles a ball of wool in *TUBB1*-RT. The healthy controls are shown in the upper pictures.

**Figure 8 jcm-09-00539-f008:**
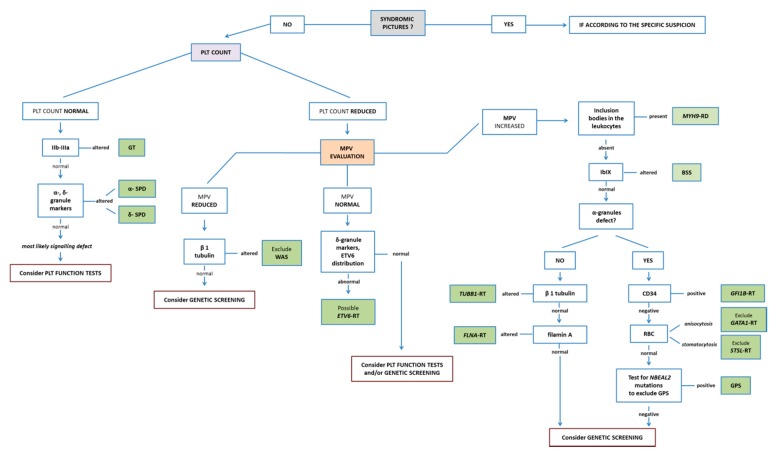
**Proposal of a diagnostic algorithm for IPDs.** According to peculiar clinical pictures (e.g., syndromic forms of IPD), or the evaluation of platelet number and size, the specific markers to be evaluated stepwise using light- and immunofluorescence microscopy are reported. Although the present diagnostic algorithm is primarily based on the evaluation of the blood smear, the use of further diagnostic tools such as platelet function tests and genetic screening should also be taken into account.

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
