# Peer review of "Diagnosis of Inherited Platelet Disorders on a Blood Smear"

_jcm, 2020, doi:10.3390/jcm9020539_

Round 1

Reviewer 1 Report

This is a very comprehensive manuscript detailing the detection of platelet abnormalities from a blood smear with light and fluorescence microscopy.  The knowledge of the authors is sound and the images excellent. The interest to a general medical audience will be limited but to a Haematology audience will be substantial.

Due to English not being the first language of the authors there is a lot of correcting required to the text. In some places it is difficult to understand because of the grammar used and the choice of words. The following are suggested corrections. Unfortunately the time that this is taking is too long for me to complete in the time I have been given to undertake this review and with my other commitments.

Line 40: Change 'coming' to 'development' and 'is enriched' to 'has become'

Line 41: Change 'fast' to 'rapidly'.

Line 46: Insert comma space 'such' before 'as RUNX1' and a comma after ETV6

Line 47: remove comma after precursors

Line 48: insert a comma after 'proteins'

Line 49 insert a comma after 'filamin A'

Line 66: insert 'a' before 'few' and replace 'additionally' with 'additional'

Line 67: replace 'alteration of' with 'compromised'

Line 90: replace 'one fourth' with '25%' and 'one third' with '33%'

Line 96: Replace the sentence with: 'Haematopoietic stem cell transplantation (HSCT) is indicated for CAMT, WAS, severe phenotypes or poor prognosis IPDs, otherwise treatment is mostly symptomatic'.

Line 100: Replace with 'For the therapeutic management of moderate haemorrhage, antifibrinolytic agents and desmopressin are effective options.'

Line 101: Insert: 'Only in the case...'

Line 102: Insert: '...compounds, such as recombinant...' 

Line 107: Replace 'delivery' with 'childbirth'

Line 109: delete 'even'

Line 110: Correct: Glanzmann's

Line 112: change 'his' to 'their' and change 'the' TPO-receptor to 'a' TPO-receptor.

Line 113: delete 'Eltrombopag' as this is not the only TPO-RA

Line 114: Replace with the following: 'Splenectomy should not, however, be adopted in patients who are candidates for HSCT as it worsens the disease-related immunodeficiency [54].'

Line 117: Insert  '..MYH9-RD, such as...'

Line 118: Replace 'contrast' with 'overcome'

Line 121: Replace with the following: 'IPDs are suspected when there is an unusual bleeding history and/or other phenotypic clues are present.'

Line 123: Insert and correct '...can be made relatively easily.'

Line 127: replace 'record' with 'documentation'

Line 128: insert '...tool, such as the ISTH...'

Line 128: Correct 'fist' to 'first'

Line 129: Change 'comprehends' to 'requires an'

Line 131: insert commas before and after 'in the first instance'

Line 132: replace 'The' with 'These'

Line 133: Replace 'various drawbacks' with 'several shortcomings'

Line 133: Insert 'level of standardisation of such assays is low.'

Line 135: Replace 'in place' with 'at the place of analysis'.

Line 135: Delete 'Mainly' and start with 'For'

Line 135: delete 'content of platelets' and replace with 'platelet concentration'

Line 136: replace 'to be analysed' with 'for effective analysis'

Line 137: Deplete 'Plus' and replace with 'Additionally,'

Line 138: Replace with 'the collection of large volumes of blood is not possible from neonates and young children'

Line 139: Delete 'Once used as a confirmation test to few disorders' and start with 'Genetic analysis has taken...'

Line 141: replace 'unravel' with 'the unravelling of'

Line 141: replace with '..IPD, and thus immediately became a significant diagnostic tool in the field.' Delete 'at once'

Line 143: Replace 'As becoming cheaper, one may imagine that NGS could trump all others diagnostic techniques' with 'As it becomes cheaper to perform, it is not inconceivable that NGS could become the standard above other diagnostic techniques'.

Line 144: Delete 'Genetic testing is particularly relevant for genetic counselling and prenatal diagnosis'

Line 149 & 150: Replace 'one half' and 'one fourth' with '50%' and '25%'

Lin 151: Replace 'probability to achieve a definite diagnosis' with 'probability of achieving a definitive diagnosis'

Line 153: Replace 'can currently' with 'may'

Line 156: Replace 'either' with 'both'

Line 156: Add after 'power' 'for the following reasons:' and delete the full stop

Line 157: Correct to 'It requires a low volume of blood (....), thus is easily collected even from newborns. Secondly, samples can be shipped safely....

Line 158: Change to 'Thirdly, although'

Line 161: Replace 'Conversely' with 'Fourthly'

Line 167: Correct to: '...during interventions, as in the case of...'

Line 169: Correct to: 'By labelling a few proteins, a potential abnormality can be identified or narrowed-down to a group of IPDs...'

Line 172: Correct to: '..(GFI1B-RT), in the case of large...'

Line 172: Delete 'are present'

Line 176: Replace 'this approach' with 'double-stained blood smear analysis'

Line 177: Replace 'a percentage around' with 'approximately'

Line 178: Replace 'substantially comparable' with 'equivalent'

In the above statement, you should provide the results that were obtained in the previous major case series. How close was it to 30%?

The Technical Considerations section should start with the details on the blood collection prior to describing the blood smear process. Suggest transfer lines 196 and most of line 202 to the start of the blood smear section.

The remainder of line 202 should stay after the YouTube link commencing a new paragraph and starting with 'Artefactual...

Line 205: Replace 'recommendable' with 'recommended'

Line 206: Replace 'by each patient' to 'from each patient'

Line 206: Insert '..only 2 colors per slide (red...'

Figure 1: It would appear that the smear direction for D is in the opposite direction to A-C. It would be better that the smear direction is the same for all slides. Does this image come out better if the slide are put on black paper? The oval-shaped tail is difficult to see.

Line 214: Replace '...and analysis is usually performed in specialised...'

Line 216: replace 'planning' with 'plan'

Line 217: replace 'by' with 'using'

Line 223: Replace 'directly-labelled' with 'directly conjugated'

Line 224: replace 'that bind to primary ones' with 'that bind to the primary antibodies'

Line 227: delete 'as'

Line 233: replace 'proof' with 'prove'

Line 234: Insert 'In the case...'

Line 235: Replace 'can' with 'may'

Line 235: Replace 'despite the antibody does not bind' with 'despite the antibody not binding'.

Line 238: insert '..Wright-stained blood film...'

Line 250: Replace 'MPV' with 'MPD'

Line 255: Rewrite 'determines the typical eosinophilic aspect'. Not sure what you are trying to say but neutrophils and NK cells also contain azurophilic granules.

Line 257: Replace 'with' to 'and'

Line 258: Delete 'pretty'

Line 260: Replace 'As regards' with 'Regarding'

Line 260: Insert '..only in the case...'

Line 260: Change to '..severe reduction of α-granules will platelets appear....

Line 264: Change to '..despite the best efforts to avoid platelet activation...'

Line 268: Insert: 'can be observed in the platelets...'

Line 269: Insert: '...is the use...'

Line 270: What are you referring to with 'capillary blood'? Blood from a capillary blood vessel or blood collected through a glass capillary tube?

Line 270: Replace with the following: '..or the time taken between dropping the 3 uL of blood onto the slide and creating the smear was too long resulting in the activation of platelets through contact with the glass surface.'

Author Response

Reviewer number 1

Response: We are extremely grateful to this reviewer for helping us so much. This is unusual and highly appreciated. We have followed all editorial suggestions.

In the above statement, you should provide the results that were obtained in the previous major case series. How close was it to 30%?

Response: Thank you. We now specified in the text the precise percentage.

The Technical Considerations section should start with the details on the blood collection prior to describing the blood smear process. Suggest transfer lines 196 and most of line 202 to the start of the blood smear section.

Response: Thank you. We are referring to a capillary blood vessel. We modified the text accordingly.

Figure 1: It would appear that the smear direction for D is in the opposite direction to A-C. It would be better that the smear direction is the same for all slides. Does this image come out better if the slide are put on black paper? The oval-shaped tail is difficult to see.  

Response: Thank you for the suggestion. Unfortunately, a black background did not give better visibility of the tail. We now turned the slide number 4, so that all the tails are in the same direction.

Reviewer 2 Report

This manuscript is a nice and comprehensive outline of the diagnostic management of inherited platelet disorders. However, there is a couple of suggestions:  

Please revisit your abbreviations. For instance, P-selectin is abbreviated as PS. In hemostasis field, PS is primarily used for protein S, and in later years for phosphatidylserine. CD-62p or P-sel would be more appropriate in this case. Another example: GT is not a standard abbreviation for Glanzmann thrombasthenia. There are a few others. Please check them all.  You may consider restructuring your manuscript according the algorithm you presented in Figure 8.

Author Response

Reviewer number 2

This manuscript is a nice and comprehensive outline of the diagnostic management of inherited platelet disorders. However, there is a couple of suggestions: 

Please revisit your abbreviations. For instance, P-selectin is abbreviated as PS. In hemostasis field, PS is primarily used for protein S, and in later years for phosphatidylserine. CD-62p or P-sel would be more appropriate in this case.  Another example: GT is not a standard abbreviation for Glanzmann thrombasthenia. There are a few others. Please check them all. 

Response: Thank you. We checked and revisited abbreviations in the text for better readability.

You may consider restructuring your manuscript according the algorithm you presented in Figure 8.

Response: This is a good suggestion, but when we tried to change the order we got into trouble with some logic flow of the manuscript. We adapted some passages to make it better readable.